# Dissection of a rice *OsMac1* mRNA 5' UTR to uncover regulatory elements that are responsible for its efficient translation

**Hiromi Mutsuro-Aoki[1]ʘ, Hiroshi Teramura[1]ʘ, Ryoko Tamukai[1], Miho Fukui[1], Hiroaki Kusano[1]¤, Mikhail Schepetilnikov[2], Lyubov A. Ryabova[2], Hiroaki Shimada[1]***

1 Department of Biological Science and Technology, Tokyo University of Science, Katsushika, Tokyo, Japan,
2 Institut de Biologie Moléculaire des Plantes, CNRS, University of Strasbourg, Strasbourg, France

ʘ These authors contributed equally to this work.
¤ Current address: Laboratory of Plant Gene Expression, Research Institute for Sustainable Humanosphere, Kyoto University, Uji, Japan
* shimadah@rs.tus.ac.jp

**Data Availability Statement:** The accession number of OsMac1 is AP004685 and its

## Abstract

The untranslated regions (UTRs) of mRNAs are involved in many posttranscriptional regulatory pathways. The rice *OsMac1* mRNA has three splicing variants of the 5' UTR (UTRa, UTRb, and UTRc), which include a CU-rich region and three upstream open reading frames (uORFs). UTRc contains an additional 38-nt sequence, termed sp38, which acts as a strong translational enhancer of the downstream ORF; reporter analysis revealed translational efficiencies >15-fold higher with UTRc than with the other splice variants. Mutation analysis of UTRc demonstrated that an optimal sequence length of sp38, rather than its nucleotide sequence is essential for UTRc to promote efficient translation. In addition, the 5' 100 nucleotides of CU-rich region contribute to UTRc translational enhancement. Strikingly, three uORFs did not reveal their inhibitory potential within the full-length leader, whereas deletion of the 5' leader fragment preceding the leader region with uORFs nearly abolished translation. Computational prediction of UTRc structural motifs revealed stem-loop structures, termed SL1-SL4, and two regions, A and B, involved in putative intramolecular interactions. Our data suggest that SL4 binding to Region-A and base pairing between Region-B and the UTRc 3'end are critically required for translational enhancement. Since UTRc is not capable of internal initiation, we presume that the three-dimensional leader structures can allow translation of the leader downstream ORF, likely allowing the bypass of uORFs.

## Introduction

Gene expression must be regulated precisely to maintain the homeostasis of cell function. Protein synthesis—a key process in gene expression—depends on mRNA levels and is mediated by numerous protein factors and protein-RNA complexes. The translation process in eukaryotic cells is divided broadly into initiation, elongation and termination stages, with translation initiation being the most rate-limiting step in protein synthesis. Initiation of translation involves a large number of translation initiation factors (eIFs), some of which are required for the initial binding of the 43S preinitiation complex [43S PIC, composed of the 40S ribosomal subunit and several soluble initiation factors, including a ternary complex (eIF2/GTP/Met-

corresponding gene ID is Os06g072660 in RAP-DB database.

**Funding:** Grant from the Ministry of Agriculture, Forestry and Fisheries (MAFF), Japan; Genome for Agricultural Innovation [grant number IPG-0022]; and Grants-in-Aid for Scientific Research from the Ministry of Education, Culture, Sports, Science and Technology (MEXT) [grant number 21570050 to H. S]. French Agence Nationale de la Recherché [grant number BLAN2011_BSV6 010 03 and ANR-14-CE19-0007 to L.R].

**Competing interests:** No authors have competing interests.

**Abbreviations:** CAPS, cleaved amplified polymorphic sequence; GUS, β-glucuronidase; ORF, open reading frame; PCR, polymerase chain reaction.

tRNA$^{Met}$)], to the capped 5'end of the mRNA to form the 48S PIC. The 43S PIC scans the mRNA until it recognizes the first initiation codon, AUG. At this point, initiation factors are released, and the 60S ribosomal subunit joins the complex to start the elongation step of protein synthesis [1, 2]. eIF5 plays an important role in the selection of the AUG start codon, and eIF5-induced hydrolysis of eIF2-bound GTP enhances translational fidelity [3].

Protein synthesis is regulated primarily at the initiation stage by mechanisms that include modulation of the activity of translation initiation factors or sequence-specific RNA-binding proteins and microRNAs [4]. In particular, translational control is mediated by *cis*-regulatory RNA elements, located in the 5' and 3' UTRs, and their interacting *trans*-acting factors [5]. RNA *cis*-regulatory elements, such as stem-loop structures and upstream open reading frames (uORFs), generally significantly impair translation [6, 7].

The presence of uORFs might activate either leaky scanning or reinitiation mechanisms, where the latter normally downregulate translation of the downstream main ORF [8, 9]. In contrast to cap-dependent translation initiation, ribosomes can use alternative mechanisms to access the AUG start codon and bind the mRNA leader via an internal ribosome binding site (IRES) present in viral and some cellular mRNAs [10, 11]. In addition, ribosomes can bypass the central complex leader regions by nonlinear migration or ribosomal shunt, as was shown primarily for viruses [12]. In this case, ribosomes start scanning from the capped 5'end of mRNA for a certain distance, and at an upstream shunt donor site, a fraction of ribosomes is translocated to a shunt landing site on an mRNA near the 3'end of the leader, often bypassing multiple uORFs and regions of stable secondary structure.

We found that long 5' UTRs in rice *OsMac1*, *OsMac2*, and *OsMac3* mRNAs encoding putative membrane proteins significantly enhanced the efficiency of translation from the downstream ORF [13, 14]. We also found three splicing variants of the 5' UTR of *OsMac1*, one of which showed very strong translational enhancer activity that is dependent on a 38-nt fragment in the middle region of the 5' UTR (termed "sp38") [13] (Fig 1).

In this study, several *cis*-regulatory RNA elements present in *OsMac1* mRNA UTRc were identified and functionally characterized, including the 38-nt element specific for UTRc. We discuss the possible role of UTRc secondary and tertiary structures in the efficient translation of *OsMac1* mRNA.

## Materials and methods

### Construction of reporter genes

The *GUS* gene, used as a reporter, was PCR-amplified from pBI221 [15] and inserted into pENTR$^{TM}$/D-TOPO (Invitrogen) to generate pENTR-35S-GUS. The UTRc cDNA was inserted preceding the *GUS* gene in pENTR–35S–GUS to generate pENTR–35S–UTRc–GUS. Internal deletion, insertion, or substitution mutations were introduced into UTRc using the megaprimer method [16] using specific primer sets that were designed to be mutagenized. Resultant fragments containing mutant UTRcs were inserted preceding the *GUS* gene in pENTR–35S–GUS. Binary plasmids were produced using pGWB1 [17] via LR clonase reaction. The fragment for the entire ORFVII (acc no. M94887.1) was chemically synthesized and inserted into the region immediately upstream of UTRc in pENTR–35S–UTRc–GUS.

### Preparation of rice protoplasts, transformation, determination of GUS activity, and estimation of translational efficiency

Cultured rice (*Oryza sativa* L.) Oc cells were suspension-cultured in Murashige and Skoog medium [18] and used for the reporter analysis. The Oc cell line, which was established from

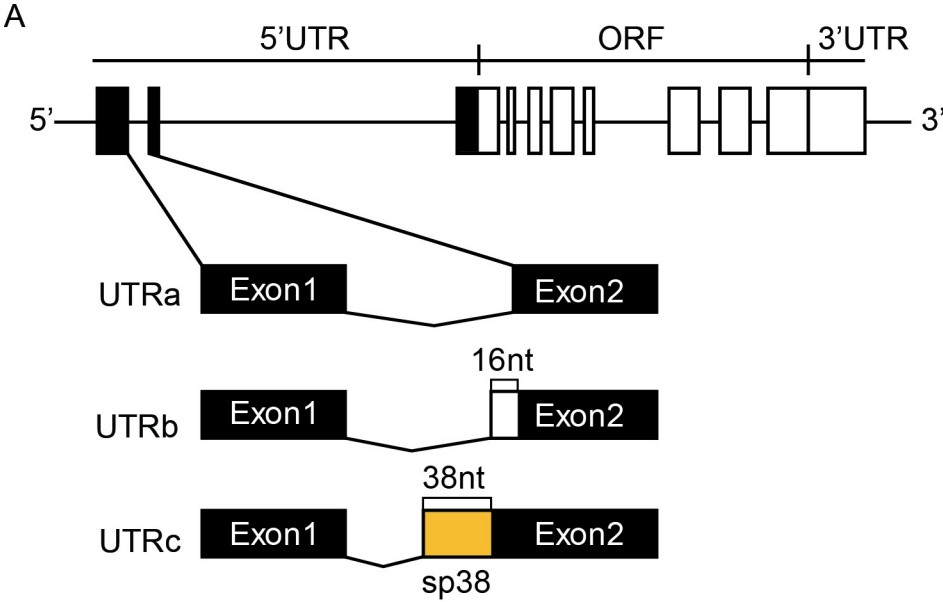

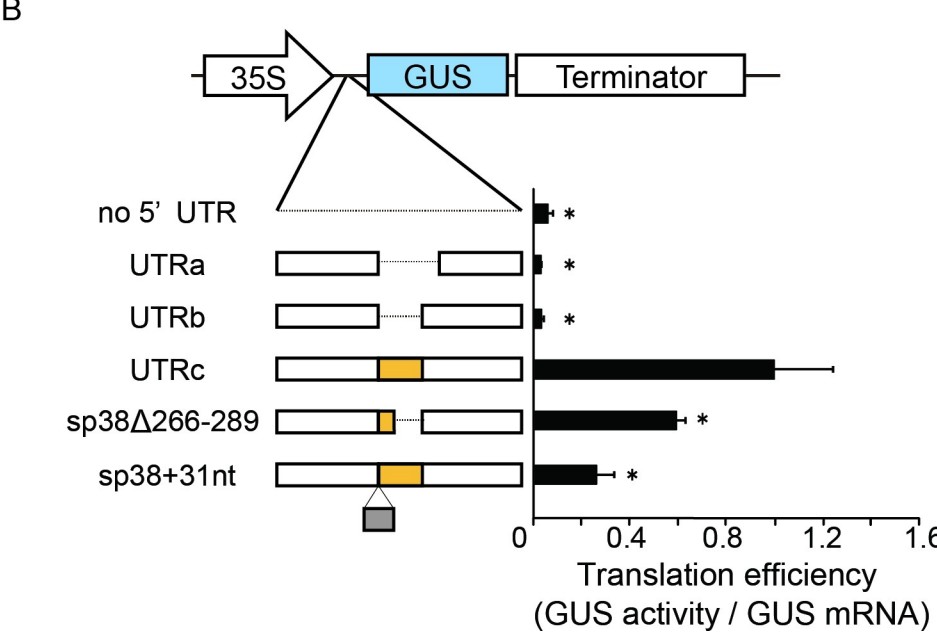

**Fig 1. Structure of OsMac1 gene and the effect of the UTRc sp38 element for the translation.** (A) Structure of splicing variants of the *OsMac1* mRNA. Boxes show the exons. Regions corresponding to the 5' UTR are indicated by filled boxes. Spliced regions are indicated by V-shaped lines. Differences among UTRa, UTRb, and UTRc are highlighted. "sp38" is shown as an orange-colored box. (B) Reporter assay of the UTRs. Protoplast of the rice suspension-cultured cells were transformed with the reporter gene containing UTRa, UTRb, and UTRc variants placed upstream of the GUS ORF (left panel). As the control, the region of untranslated sequence of *35S–GUS* configuration derived from pBI121 ("no UTR") was used. After transfection, the cells were incubated for 18 h, and GUS activities were determined. Right panel indicates the relative GUS activities normalized against the GUS mRNA, whose amount was estimated by real-time qRT–PCR. The value of the GUS activity with UTRc was set as 1.0. The nucleotide sequence of the additional 31 nt is shown in S1 Fig. The results represent the means of three independent experiments. Error bars indicate the SD (n = 3). Asterisks indicate significant differences in the translational efficiency of UTRc at $P < 0.05$.

seedling roots of indica rice accession C5924 [19, 20], was supplied by RIKEN Bioresource Center, Japan. Rice protoplast cells were prepared from cultured cells and were transformed by introduction of the desired plasmid DNA using the PEG method according to Yoo et al. (2007) [21]. The cells were incubated for 16 h at 26˚C in WI buffer (4 mM MES (pH 5.7), 0.5 M mannitol, and 20 mM KCl) and then harvested by centrifugation. The resultant protoplast cells were suspended in GUS extraction buffer (0.5 mM Tris-HCl (pH 7.0), 10 mM EDTA, 1% TritonX-100, and 1% Nonidet P-40 (Sigma-Aldrich, St. Louis, MO, USA)). The aliquots were immediately taken for *GUS* reporter gene assays. GUS activity was measured using the fluorometric assay method described in Pooggin et al. (2000) [22]. Translational efficiency was estimated as the relative value of GUS activity relative to the level of the reporter mRNA, which was determined using real-time quantitative RT-PCR. The data were analyzed statistically using Student's *t* test. In parallel, 35S–GFP [14] was introduced into protoplast cells. The amount of fluorescence signals derived from the introduced GFP gene was measured and used to monitor the efficiency of translation.

### Realtime quantitative RT–PCR analysis

Total RNA was prepared from each tissue using an RNase-free RNA preparation kit (Qiagen, Hilden, Germany) as described previously [23]. First-strand cDNA was synthesized from 1 μg of total RNA using a ReverTra Ace cDNA synthesis kit (Toyobo) with an oligo-dT (20) primer. Real-time quantitative PCR was performed as described previously [24] with SYBR Green real-time PCR mix (Toyobo) and a PCR machine. The levels of the *OsMac1* transcripts, β-glucuronidase (GUS) transcripts, and *Actin1* (AK100267) transcript were monitored by pairs of gene-specific primers: 5'-TCACATCTCCCTCAAGCTA-3' and 5'-CACGGTAGTATTCAA CTGCTTG-3', 5'-GCCGATGCAGATATTCGTA-3' and 5'-CCATCACTTCCTGATTAT TGA-3', and 5'-CCCTCCTGAAAGGAAGTACAGTGT-3' and 5'-GTCCGAAGAATTA GAAGCATTTCC-3', respectively.

## Results

### The *OsMac1* RNA UTRc leader *cis*-translation initiation element

The *OsMac1* gene (Os06g0726600) is comprised of 10 exons, where the predicted main ORF starts from a region within the third exon (Fig 1A). The corresponding cDNA consists of ~2,500 base pairs containing a long 5' UTR region comprising more than 500 nt preceding the main ORF. Several alternative splicing events were detected between the first and second exons that can lead to the formation of three variants of the *OsMac1* transcript that differ in the structure of their 5' UTR (UTRa, UTRb, and UTRc, respectively) (Fig 1A). Previously, we demonstrated that sp38 within the 5' UTRc of the mRNA of the *OsMac1* gene is critically required for translation of the leader downstream ORF [13].

    Here, we assayed the effect of the sp38 sequence and other UTRc RNA features on the translation efficiency of the ORF located downstream of the 5' UTRc. To estimate the translational efficiency of mRNAs with different 5' UTRc sequences, we constructed reporter genes carrying the β-glucuronidase (GUS) gene downstream of 5' UTR driven by the CaMV 35S-promoter (Fig 1B). These constructs were introduced into cultured rice cells to determine GUS activity and compared with that of the control construct containing the neutral 5' UTR derived from the GUS gene in pBI121 (termed 35S–GUS). The efficiency of translation of the downstream ORF for WT and mutant leader-containing mRNAs was estimated as the ratio of GUS activity to the amount of the corresponding mRNA in the cell. Cells transformed with the UTRc–GUS reporter gene exhibited a translational efficiency that was ~15-fold higher than that detected in cells harboring the control gene or transformants with UTRa–GUS or

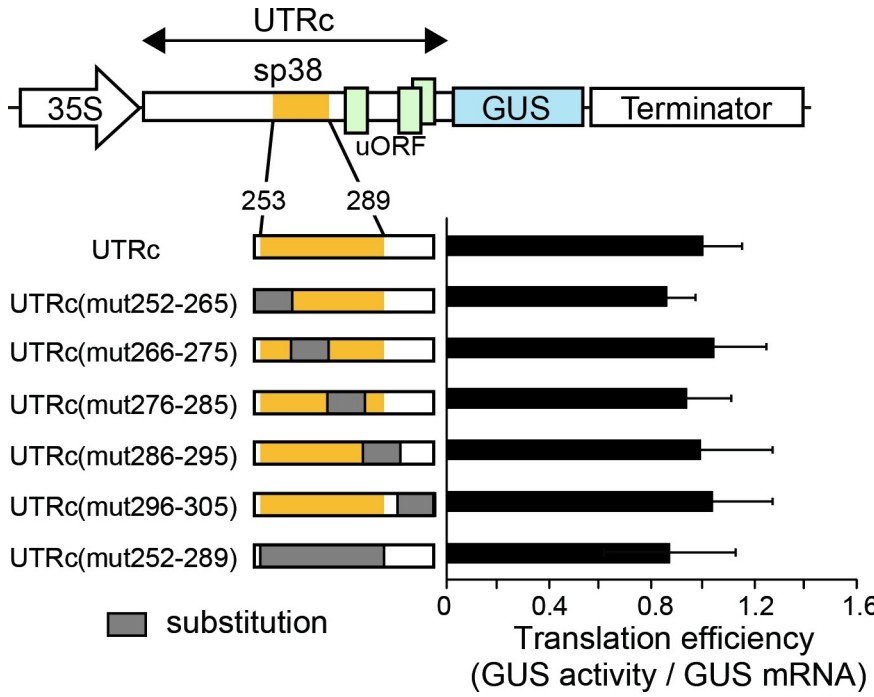

**Fig 2. Translational efficiency of the UTRc containing modified sp38 regions.** Regions of substitution in sp38 are indicated by shaded boxes (left panel). Numbers in the names of mutant genes indicate the regions of nucleotides that are substituted by the complementary ones. Right panel indicates the relative GUS activities normalized against the GUS mRNA, whose amount was estimated by real-time qRT–PCR. The value of the GUS activity with the wild-type UTRc was set as 1.0. The results represent the means of three independent experiments. Error bars indicate the SD (n = 3).

UTRb–GUS (Fig 1B). It is noted that both UTRa and UTRb lack the 38-nucleotide fragment designated sp38 (Fig 1A and S1 Fig).

To validate whether sp38 is required for the high translation efficiency of UTRc-containing mRNAs, we analyzed whether translation is sensitive to a deletion within sp38 or to an insertion event upstream of sp38. First, we generated a 23-nt deletion within the 3' region of the sp38 sequence in UTRc, constructing the reporter gene sp38(Δ266–289)–GUS; this truncated leader showed significantly reduced translational efficiency, with approximately half of the UTRc-GUS activity (Fig 1B). Strikingly, elongation of the sp38 region by 31 nt from the intron sequence preceding the sp38 fragment introduced upstream of sp38 resulted in a four-fold decrease in GUS activity (Fig 1B), suggesting that either deletion or extension of the sp38 fragment negatively affected its enhancer activity.

To determine whether the sp38 nucleotide sequence determines UTRc translation efficiency, we modified the sequence inside sp38 (nt 253–289) and the region around sp38 by replacing discrete 10–14 nt regions with their complementary nucleotides [UTRc(mut252–265)–GUS, UTRc(mut266–275)–GUS, UTRc(mut276–285)–GUS, UTRc(mut286–295)–GUS, and UTRc(mut296–305)–GUS] (Fig 2). We compared the transient expression of the reporters carrying altered sp38 sequences with the entire UTRc leader-containing reporter in cultured rice cells. As a control for transformation efficiency, a plasmid containing a single GFP ORF downstream of the TEV IRES [25] was introduced together with the indicated UTRc or mutant UTRc-containing reporter into cultured rice cells. The efficiency of translation of the downstream ORF for WT and mutant leader-containing mRNAs was estimated as the ratio of GUS activity to the GFP fluorescence produced from the corresponding control reporter.

Although values fluctuated with different nucleotide substitutions in sp38, no significant differences in translational efficiency were found. Replacement of the entire sp38 sequence with the corresponding complementary nucleotides [UTRc(mut252–289)] resulted in β-glucuronidase synthesis at comparable or slightly lower levels than with the UTRc leader (Fig 2), indicating that the nucleotide sequence within sp38 has little influence on the translational efficiency of the downstream ORF. We hypothesized that the sp38 fragment might play a role of a spacer to promote translation initiation active conformation of the 579 nucleotide 5'UTRc.

### The UTRc leader does not contain an IRES

It has been reported that some long 5' UTRs exhibit significant IRES-mediated translation [26]. To determine if UTRc contains an IRES, UTRc was placed between two ORFs (CaMV ORFVII and GUS ORF) (Fig 3). In such a bicistronic mRNA, translation of the GUS ORF is expected to be strongly inhibited. No detectable levels of GUS were produced in the presence of CaMV ORFVII upstream of UTRc (Fig 3). These results indicated that UTRc does not contain a function of IRES and that translation of the downstream ORF is dependent on the region(s) located within the long 5' UTRc.

### The three upstream open reading frames (uORFs) within the entire context of UTRc do not inhibit translation of the downstream ORF

UTRc has three uORFs (uORF1, uORF2, and uORF3) located within the 5' UTRc second half. uORF1 and uORF2 start from the nucleotide 331and 474 (Fig 4 and S2 Fig), with the AUGs of uORF1 and uORF2 residing in a moderate initiation context with R (G and A, respectively) at position –3 with respect to the first nucleotide of the start codon [27] that is recognized efficiently in plants [28]. uORF3, which has an AUG in a weak context, i.e., missing both R at

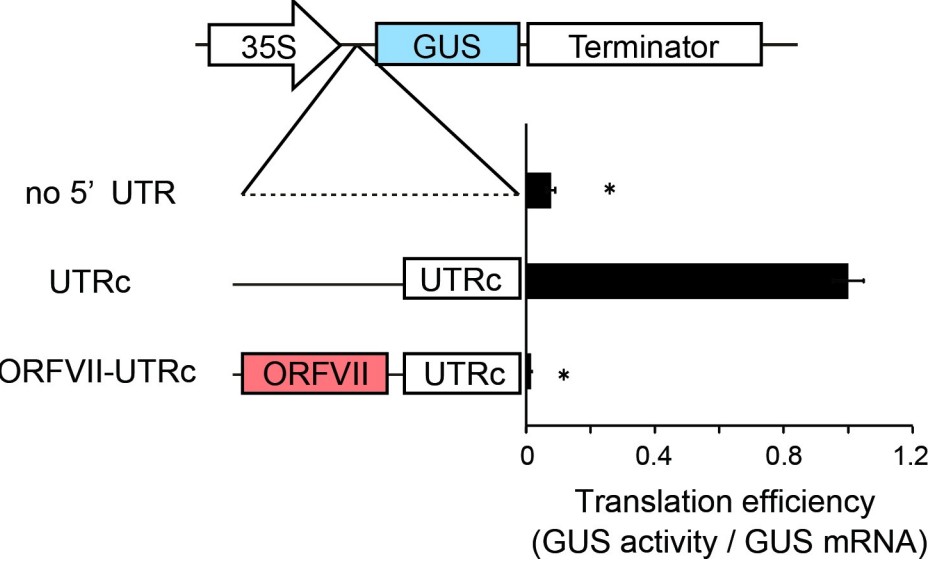

**Fig 3. Examination of the IRES function in UTRc.** "no UTR" and "UTRc" indicate 35S–GUS and UTRc–GUS, respectively. "ORFVII-UTRc" indicates the ORFVII–UTRc–GUS reporter that contains CaMV ORFVII upstream of the UTRc sequence. The results are expressed as the ratio of the enzymatic activity of GUS and the amount of corresponding GUS-containing mRNA synthesized in cultured rice cells, where the value of the GUS activity expressed from monocistronic mRNA is set as 1.0. GUS mRNA levels were estimated by real-time qRT–PCR. Error bars indicate the SD (n = 3). Asterisks indicate significant differences in the translational efficiency of UTRc at $P < 0.05$. The results shown represent the means obtained in three independent experiments.

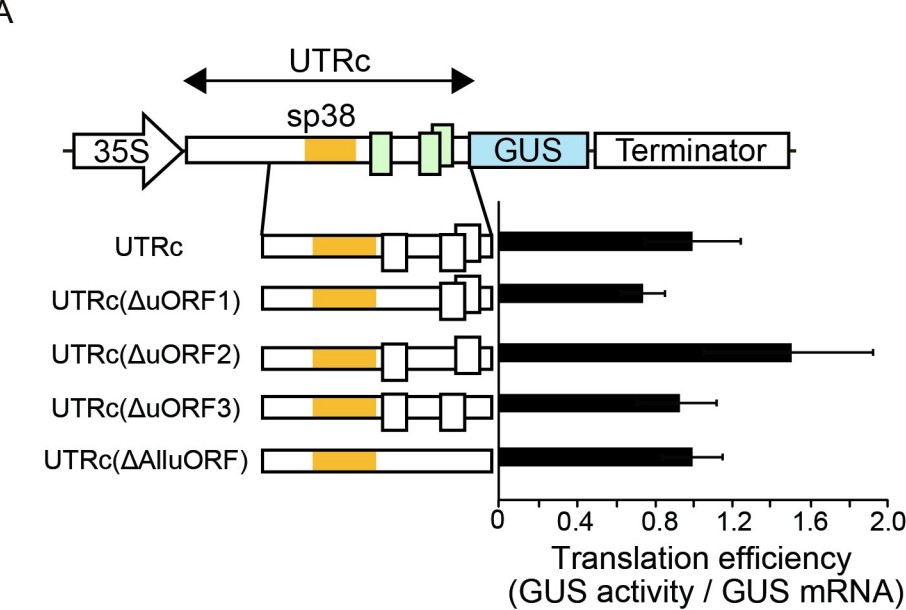

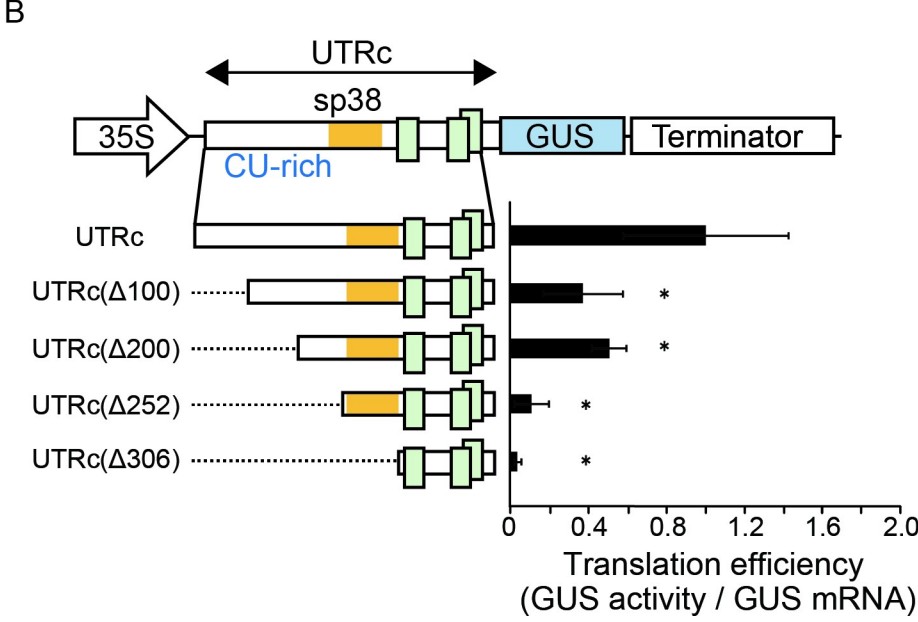

**Fig 4. Identification of regulatory UTRc *cis*-elements involved in the modulation of *OsMac1* mRNA translation.**
(A) Mutations introduced in the UTRc leader to exclude uORF1, uORF2, uORF3 or all the AUG codons (AUG codon was replaced by UUG). UTRc-, UTRc (ΔuORF1)-, UTRc (ΔuORF2)-, UTRc (ΔuORF3)-, and UTRc (ΔAlluORF)-containing reporters were used for transformations. (B) Translational efficiency of truncated UTRc mutants. UTRc (Δ100), UTRc (Δ200), UTRc (Δ252), and UTRc (Δ306) are truncated UTRc mutants in which regions of 100 nt, 200 nt, 252 nt and 306 nt were deleted from the 5' end, respectively, as shown by broken lines. sp38 and uORFs are shown as orange-colored boxes and enlarged open boxes, respectively. Translational efficiencies are shown as the relative GUS activities that are normalized by the amount of mRNA. The value of the GUS activity of UTRc is set as 1.0. The results represent the means of three independent experiments. Error bars indicate the SD (n = 3). Asterisks indicate significant differences in the translational efficiency of UTRc at $P < 0.05$.

position –3 and G at position +4, placed within the uORF2 sequence, is unlikely to be translated. Elements such as uORFs are considered to be preferential *cis*-acting translational repressors in eukaryotes [29, 30]. Translation events at either the 35-codon uORF1 and/or 25-codon uORF2 could significantly impair translation reinitiation of the main downstream ORF. Surprisingly, similar levels of GUS activity were obtained with plasmid constructs carrying the native or uORF-less leader (AUG codons were substituted by UUG codons using site-directed mutagenesis), except that a uORF2 deletion somewhat increased translation efficiency (Fig 4A), suggesting that these uORFs have no significant effect on translation of the downstream ORF in the context of the full leader. However, deletion of either 252 or 306 nucleotides at the 5' end of UTRc [UTRc(Δ252) and UTRc(Δ306)] upstream of the first uORF1 nearly abolished downstream ORF translation (Fig 4B), indicating the inhibitory capacity of uORFs if these elements are present downstream of a short leader (78 nt and 25 nt, respectively).

## The 5'-terminal 100 nt fragment of the CU-rich sequence positively impacts translation efficiency

The 5' UTRc contains a long CU-rich tract upstream of the sp38 sequence (S2 Fig). To investigate the contribution of the CU-rich region to the translational efficiency of the downstream ORF, we deleted 100, 200 or 252 nucleotides upstream of sp38 starting at the 5' end [UTRc(Δ100), UTRc(Δ200) and UTRc(Δ252), respectively] and estimated the main ORF translational efficiency in the presence of these truncated UTRs (Fig 4B). Interestingly, the translational efficiencies of both UTRc(Δ100)–GUS and UTRc(Δ200)–GUS were significantly reduced to approximately half of that observed for the wild-type construct. Furthermore, removal of the entire CU-rich region [UTRc(Δ252–GUS)] resulted in drastically lower translational efficiency, i.e., one-tenth of that observed for the full-length UTRc. Note that deletion of both the CU-rich and sp38 regions [UTRc(Δ306)–GUS] resulted in barely detectable translation from the main ORF downstream of the truncated uORF-containing leader (Fig 4B). These results demonstrate that the first 100 nucleotides of the CU-rich region positively contribute in the translation of UTRc-GUS RNA.

## Identification of UTRc secondary structure elements that improve translation initiation efficiency

Computational analysis of UTRc secondary structure revealed several conserved stem-loop structures (Fig 5). Two conserved stem-loop structures, named SL1 and SL2, were predicted at positions 15–36 nt and 107–183 nt within the CU-rich region, respectively (Fig 5). The SL3 structure can be predicted with high accuracy within the region immediately downstream of sp38; SL4, at the 3' end of the 5' UTR.

As mentioned above, deletion of the first 100 nt of UTRc [UTRc(Δ100)–GUS] with SL1 decreases UTRc translation efficiency substantially, indicating the possible involvement of SL1 in translation of the main ORF (Fig 4B). In contrast, deletion of the additional 100 nt with SL2 [the UTRc(Δ200)–GUS mutant lacks both the SL1 and the SL2 stem structures] did not result in further reduction of GUS activity, indicating that the SL2 structured region is rather dispensable for main ORF translation. SL3 folds into two stem sections separated by bifurcations, where the first stem section constitutes the most stable region of the SL3 structure (Fig 5). To test whether base pairing of this region impacts leader translation efficiency, the first stem section was destroyed by complementary mutations in the 5' arm (Fig 6A). However, the disruption of SL3 was dispensable for GUS translation, suggesting no role of SL3 in translation in the context of the full leader.

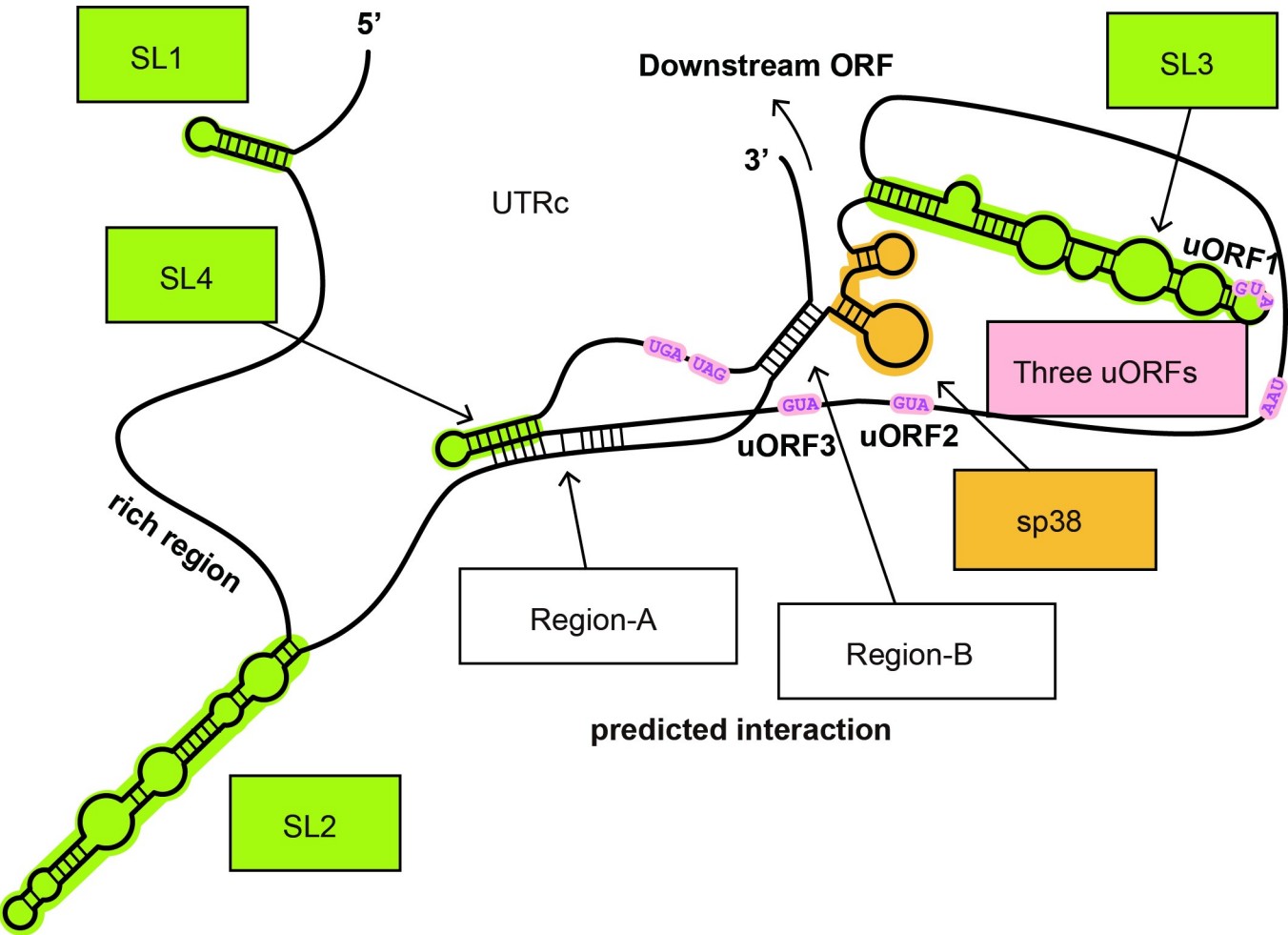

**Fig 5. Schematic representation of putative *cis*-acting RNA regulatory elements of UTRc that can impact the expression of *OsMac1* mRNA variants.** Regions of the predicted intramolecular interactions of UTRc are shown. SL1, SL2, SL3, and SL4 indicate the predicted stem-loop structures. Region–A and Region–B indicate the predicted intramolecular interactions. sp38 is colored orange. Initiation and termination codons of uORFs are colored pink. Local secondary structures of UTRb and UTRc were predicted using the CentroidFold program (http://www.ncrna.org/centroidfold). The nucleotide sequence of UTRc is shown in S3 Fig.

According to tertiary structure predictions, SL4 can base pair with the region upstream of sp38 (Region–A). To analyze the effect of the interactions within Region–A on translational efficiency, base pairing within SL4 was destroyed by deletion of the corresponding 18 nt (nt 511–529). This truncation mutation resulted in a significant reduction in translation efficiency (Fig 6B). Region–A was predicted to interact with the left arm of the SL4 secondary structure (between nt 210–224 and nt 515–502) (Fig 5). Truncation of the SL4 region can be considered as a mutant lacking interaction with Region–A. These results further indicate that although the SL4 secondary structure was not effective, a left arm sequence that can base pair with Region–A is critical for translation enhancement.

In addition, stable base pairing was predicted between the 3' end UTRc sequence (Regions–B; nt 564–559) and the CU-rich sequence just upstream of sp38 (nt 242–250) (Fig 5). To investigate the impact of this putative base-paired region on the translation of UTRc-GUS mRNA, the predicted base pairing with Region–B was destroyed by the introduction of 10 complementary mutations into Region–B (nt 242–250) (Fig 6C, upper central panel). These mutations

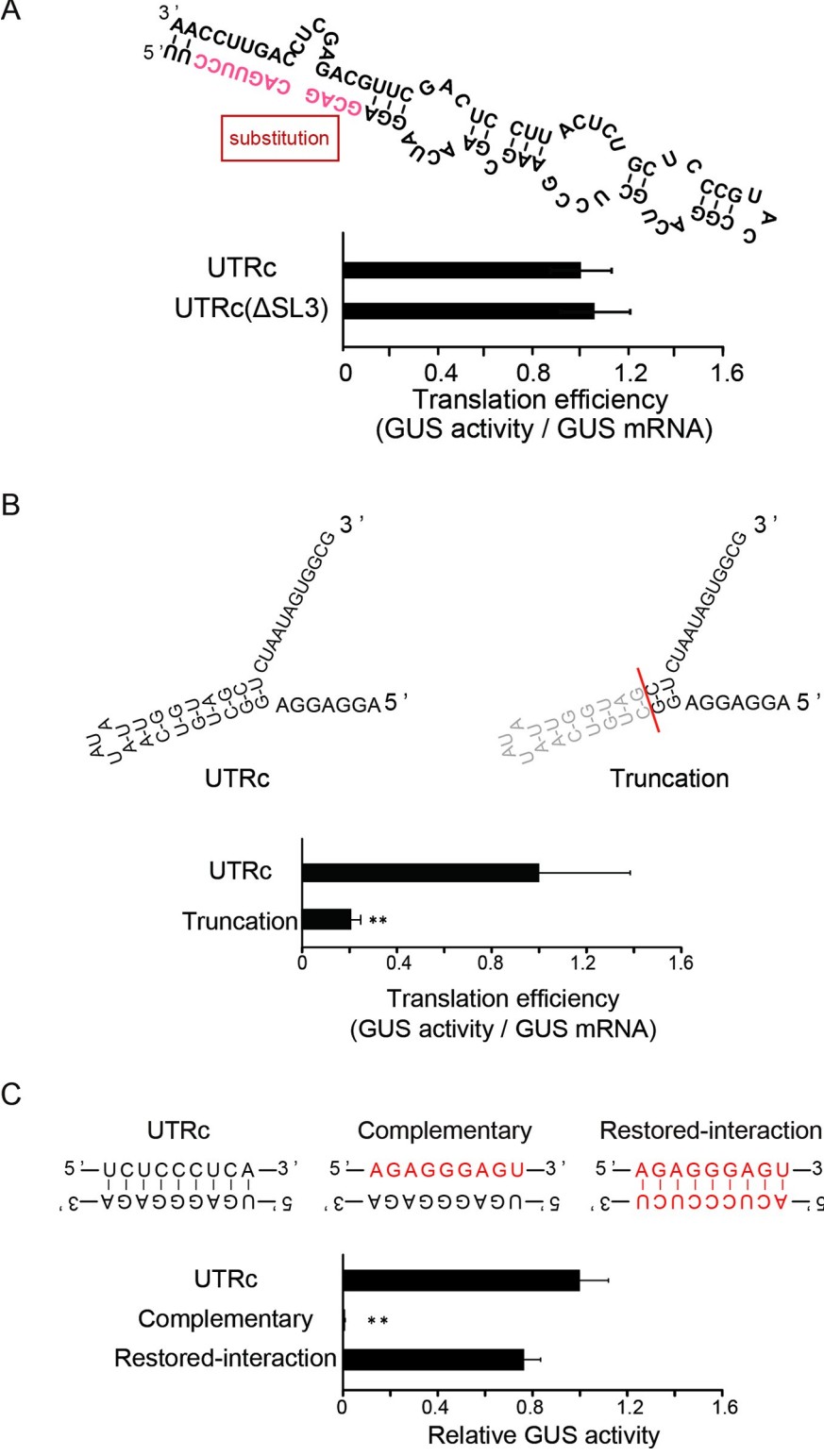

**Fig 6. Dependence of the *OsMac1* mRNA translation efficiency on secondary and tertiary UTRc structures.** (A) Disruption of SL3 stem section 1 is dispensable for UTRc function. Schematic representation of stem section 1 of SL3 (upper panel). Complementary mutations in the 5' arm of stem section 1 are shown by red letters, UTRc (ΔSL3). (B) Translational efficiency of SL4 mutants. Schematic representation of entire mutated and truncated SL4 (upper panels). "Truncation" indicates a deletion mutant lacking the stem-loop as indicated. (C) Schematic representation of a base-

paired motif between Region-B (nt 564–559) and UTRc (nt 242–250) (upper panels). Two sets of mutations are shown: those introduced as complementary to destroy base pairing ("Complementary") and the additional complementary mutations introduced to rebuild the base pairings ("Restored-interaction"). Translational efficiencies are shown as the relative GUS activities that are normalized by the amount of mRNA. The value of the GUS activity of UTRc is set as 1.0. The results represent the means of three independent experiments. Error bars indicate the SD (n = 3). Asterisks indicate significant differences in the translational efficiency of UTRc at $P < 0.05$.

resulted in a 90% reduction in translation. When the structure was restored with complementary mutations on the counterpart sequence in the mutant (UTRc, nt 242–250), the original level of GUS activity was restored (Fig 6C, upper right panel), suggesting the crucial importance of base pairing between Region–B and the 3' end portion of the leader sequence.

We confirmed the transcripts of the mutant UTRcs whose translational efficiency significantly altered. RT-PCR analysis detected the expected sizes corresponding to the mRNA, suggeting no aberrant splicing variant in these transcripts (S3 Fig).

## Discussion

Rice *OsMac1* mRNA has three variant 5' UTRs (UTRa, UTRb, and UTRc), which result from alternative splicing between the first and second exons, where only UTRc-containing mRNA is abundantly translated [13]. This suggests that the UTRc configuration harbors several RNA *cis*-elements promoting translation of the downstream ORF. In particular, the 38-nt region (sp38) unique to UTRc was suggested to play a critical role in translation initiation [13]. Our results demonstrated that the enhancer activity of sp38 was not dependent on the nucleotide sequence but was significantly influenced by differences in sp38 region length, strongly indicating that the size of the sp38 fragment is optimal for UTRc enhancement (Fig 1). Future analysis will be required to determine whether the sequence length of this region is the only factor responsible for the variation in translation efficiency. Although UTRs a, b and c contain a CU-rich region, three conserved uORFs and four regions predicted to form stem–loop structures (SL1 to SL4; Fig 5), UTRc tertiary structure predictions indicated the existence of intramolecular interactions between SL4 and Region–A and strong base pairing between Region–B and the UTRc 3' end that differed from those predicted for UTRa and UTRb.

Some mRNAs contain an unusual polypyrimidine tract at their transcriptional start site that is involved in translational regulation [31, 32]. It has been reported that RNA-binding and polypyrimidine tract-binding proteins promote translation by binding to RNA motifs that are predicted to form specific secondary structures [33, 34]. Here, truncation of the first 100 nt of the CU-rich region that predicted to contain a stem-loop structure resulted in significant reduction in translational efficiency. Further analysis would require to determine whether the CU-richness, or structure, or protein binding of the 100 nt 5'UTR fragment is important for high leader activity (Fig 4B).

uORFs are known as regulators of translation [29, 30, 35–37]. Translation of uORF-containing mRNAs depends on a reinitiation mechanism, where ribosomes terminating translation of a uORF resume scanning and reinitiate at an ORF further downstream on the same mRNA. Reinitiation is strictly regulated in eukaryotes by the length of the uORF (< 25 codons) [38]. Generally, uORFs are considered as repressors of translation of ORFs downstream of the leader [28]. The *OsMac1* mRNA harbors three uORFs encoding 35, 25, and 23 amino acids, respectively, suggesting that these uORFs can inhibit downstream translation. Mutating uORF start codons separately or together led to no significant alteration of translational efficiency in the context of the entire leader (Fig 4A). In contrast, removal of 252 or 306 nucleotides upstream of the first uORF nearly abolished translation of the main ORF (Fig 4B). Thus, it is presumed that the retention of either 78 or 25 nt of 5' UTR upstream of the first

uORF is inhibitory for translation of the main ORF suggesting strong inhibitory potential of uORFs.

Interestingly, our results strongly suggest that the length of the sp38 spacer and Region–A and Region–B, both of which are involved in intramolecular interactions with the leader 3' end within UTRc, are critical for efficient translation. Indeed, SL4 is located in the 3' region of UTRc, upstream of uORF AUGs, and may form base-pairings with Region–A (Fig 5). We predicted and confirmed the functional role of another base-paired element between Region–B and the 3' leader sequence located further downstream of SL4. The importance of the latter intramolecular interactions was confirmed by mutations and second site reversions of this structure, which abolished and reestablished interactions, respectively (Fig 6). It can be speculated that the formation of both structural elements can pause scanning ribosomes and promote their bypass of the central region with three uORFs and initiate upstream of the main ORF start codon. A ribosomal shunting mechanism has been proposed to explain how the translation machinery can overcome the obstacles imposed by the cellular inhibitor of apoptosis protein 2 (cIAP2) mRNA leader, which folds into a unique secondary structure and contains a high number of uORFs [39]. The cIAP2 5' UTRc forms unique and stable stem structures that might trigger scanning ribosomes to bypass uORFs via ribosome shunting. Similar scenario we can envisage for translation initiation of ORF downstream of the UTRc leader, but this would require further research.

Multiple models of cap-dependent or cap-independent mechanisms of translation initiation have been described in plants [26, 40–42], where some viral transcripts contain IRES sequences composed of several hundred nucleotides that recruit the small subunit of the ribosome to initiate eukaryotic translation internally [43]. In the case of UTRc, we detected no IRES-dependent internal initiation (Fig 3). However, the intramolecular interactions formed within UTRc play an important role in translational enhancement and may promote bypass of the inhibitory leader regions that contain uORFs by ribosomes scanning from the 5' end of the mRNA, directing the landing of scanning ribosomes just upstream of the main ORF.

We used the vector system driven by the 35S promoter, which may provide a large amount of reporter transcript. The translational enhancement may occur in a context-dependent manner. It would be necessary to determine the translational efficiency using a different promoter system. Although we cannot exclude the existence of as yet unidentified factors promoting translational efficiency, their identification would require further research.

## Supporting information

**S1 Fig.** (A) Nucleotide sequence of the control 5' UTR, which is derived from pBI121 (AF485783). Numbers indicate the nucleotide position in the plasmid shown in the database. Regions for CaMV 35S promoter and the coding sequences of gusA gene are indicated by green and blue boxes, respectively. (B) Nucleotide sequences of the region around sp38. UTRc +31nt contains an additional 31 nt sequence derived from the 3' portion of the intron 1. Gaps are introduced into the region lacking in each UTRs. sp38 in UTRc is indicated by an orange box.
(PDF)

**S2 Fig. Nucleotide sequence of UTRc.** The CU-rich sequence in the 5' region is indicated by blue letters. Nucleotide sequence of sp38 is shown in an orange box. Splicing sites in UTRa and UTRb are indicated by red triangles. uORFs are shown in red boxes. Initiation codons AUG are indicated by red letters. The downstream ORF of OsMac1 is shown in a yellow box. Nucleotide sequences consisting of Region–A and Regions–B are double-underlined.
(PDF)

**S3 Fig. Analysis of integrity of the reporter mRNA during rice suspension culture proto-plast incubation.** mRNA for the reporters were obtained by RT-PCR immediately after testing their translation efficiencies. cDNA was amplified using primers corresponding the entire 5'UTR region and the GUS coding region. From left to right: cDNA (UTRc) contains the control DNA fragment amplified from the wild-type UTRc; UTRc indicates the transcript amplified from the wild-type UTRc construct; "Complementary", "Restored-interaction", and "Tuncation" shows corresponding mutant transcripts described in Fig 6. A Gap is introduced between the lanes of the same gel. Size markers used are 100 bp DNA ladder marker (New England Biolabs, Ipswich, MA, USA).
(PDF)

**S1 Raw images.**
(PDF)

## Author Contributions

**Conceptualization:** Hiroaki Shimada.

**Formal analysis:** Hiromi Mutsuro-Aoki, Hiroaki Kusano.

**Investigation:** Hiromi Mutsuro-Aoki, Hiroshi Teramura, Ryoko Tamukai, Miho Fukui, Hiroaki Kusano, Mikhail Schepetilnikov.

**Methodology:** Mikhail Schepetilnikov, Lyubov A. Ryabova.

**Supervision:** Lyubov A. Ryabova, Hiroaki Shimada.

**Validation:** Mikhail Schepetilnikov, Hiroaki Shimada.

**Writing – original draft:** Hiromi Mutsuro-Aoki, Lyubov A. Ryabova, Hiroaki Shimada.

**Writing – review & editing:** Hiroaki Kusano, Mikhail Schepetilnikov, Lyubov A. Ryabova, Hiroaki Shimada.

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
