## [Decision Letter · Decision Letter 0]

14 Apr 2021

PONE-D-21-09082

Dissection of a rice OsMac1 mRNA 5' UTR to uncover regulatory elements that are responsible for its efficient translation

PLOS ONE

Dear Dr. Shimada,

Thank you for submitting your manuscript to PLOS ONE. After careful consideration, we feel that it has merit but does not fully meet PLOS ONE’s publication criteria as it currently stands. Therefore, we invite you to submit a revised version of the manuscript that addresses the points raised during the review process.

　I have asked two experts for review your manuscript. Both of them found your manuscript contains important information although need further improvement. Please follow the suggestions by two reviewers and reply all questions and suggestions.

We look forward to receiving your revised manuscript.

Kind regards,

Minami Matsui

Academic Editor

PLOS ONE

Journal Requirements:

"This work was supported by a Grant from the Ministry of Agriculture, Forestry and Fisheries

(MAFF), Japan; Genome for Agricultural Innovation [grant number IPG-0022]; and

Grants-in-Aid for Scientific Research from the Ministry of Education, Culture, Sports, Science

and Technology (MEXT) [grant number 20770037 to T. S., grant number 21570050 to H. S].

This work was supported by French Agence Nationale de la Recherché—BLAN-2011_BSV6

010 03 and ANR-14-CE19-0007—funding to L.R."

"The authors received no specific funding for this work."

Reviewers' comments:

Reviewer's Responses to Questions

**Comments to the Author**

1. Is the manuscript technically sound, and do the data support the conclusions?

Reviewer #1: Yes

Reviewer #2: Yes

2. Has the statistical analysis been performed appropriately and rigorously? 

Reviewer #1: Yes

Reviewer #2: Yes

3. Have the authors made all data underlying the findings in their manuscript fully available?

Reviewer #1: Yes

Reviewer #2: No

4. Is the manuscript presented in an intelligible fashion and written in standard English?

Reviewer #1: Yes

Reviewer #2: Yes

5. Review Comments to the Author

Reviewer #1: This is a follow-up article after Teramura et al (2012, in the reference list). In the previous work, they found translational enhancement in one of the three types of transcripts of OsMac1 caused by alternative splicing, and that a 38 nt sequence specific to the variant, UTRc, is indispensable for the enhancement. The submitted report tried fine mapping of the corresponding elements possibly within the 38 nt region.

Using transient assays with rice protoplasts, they found that the 38 nt functions in a sequence-independent manner and its length is important. Then the authors analyzed outside of the 38 nt region. They found there are several possible secondary structures in the 5' UTR regions as shown in Fig. 5. Mutation and deletion analyses revealed pairing of Region-B is critical for the translational enhancement in addition to sp38. These clearly identified elements appear to be helper elements because both do not show sequence specificity (Fig. 2 & 6C). Therefore, this report could not identify the key element for the translational enhancement in UTRc of OsMac1.

Currently known molecular model for translational enhancement by 5' UTR is enhancement of ribosome binding by an attractor sequence within 5' UTR, providing so-called internal ribosomal entry which is independent of the cap. This is the only one molecular model for translational enhancement. Interestingly, results shown in Fig. 3 suggest this model does not fit the case of UTRc. I think this finding is very important, because studies of this case (UTRc) would lead to a novel molecular model for translational enhancement.

Because this report has important results in the field of translation study, not only in plant science but in general molecular biology, I recommend its publication in PLoS ONE.

Major points :

none

Minor points:

Fig. 1B bottom pictures: they are the same as ones in a published report (Fig. 4B, Teramura et al, Plant Biotechnol 29: 43-49, 2012). They should be changed to original ones.

Previous article in 2012 and this report both use the same vector system with 35S promoter/reporter/TNOS. Some of the translational enhancement occurs in a context-dependent manner. Context means utilized 5' UTR end from the promoter seq, CDS, and 3' UTR. The translational enhancement by UTRc was confirmed using two different reporters (GUS and GFP), but the same promoter and 3' UTR were used throughout the studies. It would be necessary to confirm context-independence of the phenomenon in the future studies.

Reviewer #2: The manuscript titled “Dissection of a rice OsMac1 mRNA 5' UTR to uncover regulatory elements that are responsible for its efficient translation" by H. Mutsuro-Aoki, H. Teramura, R. Tamukai, M. Fukui, H. Kusano, M. Schepetilnikov, L.A. Ryabova, and H. Shimada, is an important piece of work, casting light on the molecular comprehension of possible roles of UTRc secondary and tertiary structures in the translation efficiency of mRNAs.

The manuscript founds itself on previous research on OsMac1 published in 2012 (H. Teramura et al. 2012. “A long 5' UTR of the rice OsMac1 mRNA enabling the sufficient translation of the downstream ORF.” Plant Biotechnol. 29: 43-9.). In the current manuscript, the authors identified and functionally characterized cis-regulatory RNA elements present in OsMac1 mRNA UTRc, and proposed possible roles of UTRc secondary and tertiary structures for the efficient translation of OsMac1 mRNA.

The argumentation is well organized and based on solid data. Having that said, yet there are concerns in the manuscript that needs to be improved.

The following expression in the manuscript must be revised to avoid confusion and improve the quality.

1) P2 L5, P5 L3: Abstract describes “contains an additional 38-nt sequence, termed sp38,”, while the expression “38-nt fragment” and “38-nt element” appear in P4. Furthermore the authors will have to wait till the expression “a 38-nt sequence (termed "sp38") within the 5' UTRc” that appears in P5, for a definition. This is confusing from two points. One; it is not clear what has been discovered before and what is novel here. Two; when and how this 38-nt element is defined or renamed as sp38. Since, Teramura et al. 2012 has described this element as “additional 38-nt sequence found in UTRc” throughout their text, it is logical to clearly re-define this “additional 38-nt sequence found in UTRc” as sp38 at an early stage in this manuscript, to avoid confusion. This must be revised.

2) P5 L24: “truncated leader showed significantly reduced translational efficiency, with approximately half of the UTRc-GUS activity (Fig. 1B). Strikingly, ...” Lower panel should also show GUS staining in the callus cells harboring the reporter gene sp38(∆266–289)–GUS and sp38+31nt to assure the staining quality.

3) P6 L5: “UTRc(mut252–265)” It is not clear what number corresponds to the sp38. The number should be consistent within the manuscript. For example, numbers should be added in the bar illustrations of Fig. 2 and Fig, 4. Supplementary Fig. S2 shows sp38 to start from #253 while mutations for constructs shown in Fig. 2 start with #252 with no explanation. Furthermore the confusion is deepened by the fact that Teramura et al. 2012 defines the sp38 sequence to start from #254 in their Figure1. This contradiction should be solved or explained.

4) P7 L8: “starting from the nucleotide 330” Does this indicate that the first ORF starts from #330, whereas the first AUG is at #331? Alternatively does this indicate that the second half of UTRc starts from #330? The expression is misleading and must be revised.

5) P9 L10: “deletion of the corresponding 18 nt” This should be written out in numbers.

6) Figures: The resolution of the figures should be improved (ex, Fig. 1, Fig 5, Fig. 6).

7) Typos: Supplementary Fig. S2. has “Fig4” written on the bottom.

8) Supplementary Fig. S3.: The right gel seems to be a image of two separately run gels. The image can be used to show single bands but cannot be used to describe the length difference noted in the text due to the truncation. The data should be revised for this purpose.

6. PLOS authors have the option to publish the peer review history of their article (what does this mean?). If published, this will include your full peer review and any attached files.

Reviewer #1: **Yes: **Yoshiharu Y. Yamamoto

Reviewer #2: No

---

## [Author Response · Author response to Decision Letter 0]

24 May 2021

Response to the Reviewers 

For the revision, the manuscript was arranged to follow the PLOS ONE formant. In the file labeled 'Manuscript', the orders of sections are changed according to the instruction manual. We also uploaded a file showing the points of revision based on the original-formatted manuscript as the file labeled 'Revised Manuscript with Track Changes' that indicates the points of revisions written in the 'Response to Reviewers'. In Revised Manuscript with Track Changes, added and deleted words are indicated by red and blue letters, respectively. According to the requirement on the original image data for the gels shown in the Supplementary Fig. S3, we uploaded a PDF file which contains the image data of agarose gel electrophoresis written in the lab notebook.

To the Reviewer #1: 

This is a follow-up article after Teramura et al (2012, in the reference list). In the previous work, they found translational enhancement in one of the three types of transcripts of OsMac1 caused by alternative splicing, and that a 38 nt sequence specific to the variant, UTRc, is indispensable for the enhancement. The submitted report tried fine mapping of the corresponding elements possibly within the 38 nt region.

Using transient assays with rice protoplasts, they found that the 38 nt functions in a sequence-independent manner and its length is important. Then the authors analyzed outside of the 38 nt region. They found there are several possible secondary structures in the 5' UTR regions as shown in Fig. 5. Mutation and deletion analyses revealed pairing of Region-B is critical for the translational enhancement in addition to sp38. These clearly identified elements appear to be helper elements because both do not show sequence specificity (Fig. 2 & 6C). Therefore, this report could not identify the key element for the translational enhancement in UTRc of OsMac1.

Currently known molecular model for translational enhancement by 5' UTR is enhancement of ribosome binding by an attractor sequence within 5' UTR, providing so-called internal ribosomal entry which is independent of the cap. This is the only one molecular model for translational enhancement. Interestingly, results shown in Fig. 3 suggest this model does not fit the case of UTRc. I think this finding is very important, because studies of this case (UTRc) would lead to a novel molecular model for translational enhancement.

Because this report has important results in the field of translation study, not only in plant science but in general molecular biology, I recommend its publication in PLoS ONE.

Major points :

None

Minor points:

(1) Fig. 1B bottom pictures: they are the same as ones in a published report (Fig. 4B, Teramura et al, Plant Biotechnol 29: 43-49, 2012). They should be changed to original ones.

¬¬–––Thank you for your comments. Lower and upper panels of Fig. 1B show the qualitative and quantitative data of GUS activities, respectively. Both panels similarly show the activities of the reporter GUS protein. The data in the upper panel is more accurate than those in the lower panel. Because a result of staining has been reported in the previous paper, the lower panel of Fig. 1B was deleted and corresponding figure legend are revised (lines 481–483).

(2) Previous article in 2012 and this report both use the same vector system with 35S promoter/reporter/TNOS. Some of the translational enhancement occurs in a context-dependent manner. Context means utilized 5' UTR end from the promoter seq, CDS, and 3' UTR. The translational enhancement by UTRc was confirmed using two different reporters (GUS and GFP), but the same promoter and 3' UTR were used throughout the studies. It would be necessary to confirm context-independence of the phenomenon in the future studies.

–––Thank you for your comments. We used a simplified system using the same promoter and reporter gene in order to easily evaluate the effect of translation. We used the vector system driven by 35S promoter, which may provide a large amount of reporter transcript, and this was determined by qRT-PCR. However, it is possible that the translational enhancement may occur in a context-dependent manner. It would be necessary to determine the translational efficiency using a different promoter system. This matter was added in the Discussion (lines 292–295).

To the Reviewer #2: 

The manuscript titled “Dissection of a rice OsMac1 mRNA 5' UTR to uncover regulatory elements that are responsible for its efficient translation" by H. Mutsuro-Aoki, H. Teramura, R. Tamukai, M. Fukui, H. Kusano, M. Schepetilnikov, L.A. Ryabova, and H. Shimada, is an important piece of work, casting light on the molecular comprehension of possible roles of UTRc secondary and tertiary structures in the translation efficiency of mRNAs.

The manuscript founds itself on previous research on OsMac1 published in 2012 (H. Teramura et al. 2012. “A long 5' UTR of the rice OsMac1 mRNA enabling the sufficient translation of the downstream ORF.” Plant Biotechnol. 29: 43-9.). In the current manuscript, the authors identified and functionally characterized cis-regulatory RNA elements present in OsMac1 mRNA UTRc, and proposed possible roles of UTRc secondary and tertiary structures for the efficient translation of OsMac1 mRNA.

The argumentation is well organized and based on solid data. Having that said, yet there are concerns in the manuscript that needs to be improved.

The following expression in the manuscript must be revised to avoid confusion and improve the quality.

1) P2 L5, P5 L3: Abstract describes “contains an additional 38-nt sequence, termed sp38,”, while the expression “38-nt fragment” and “38-nt element” appear in P4. Furthermore the authors will have to wait till the expression “a 38-nt sequence (termed "sp38") within the 5' UTRc” that appears in P5, for a definition. This is confusing from two points. One; it is not clear what has been discovered before and what is novel here. Two; when and how this 38-nt element is defined or renamed as sp38. Since, Teramura et al. 2012 has described this element as “additional 38-nt sequence found in UTRc” throughout their text, it is logical to clearly re-define this “additional 38-nt sequence found in UTRc” as sp38 at an early stage in this manuscript, to avoid confusion. This must be revised.

––– According to the reviewer’s suggestion, sp38 was defined in Introduction (line 82), and the second one was described by sp38 (line 98).

2) P5 L24: “truncated leader showed significantly reduced translational efficiency, with approximately half of the UTRc-GUS activity (Fig. 1B). Strikingly, ...” Lower panel should also show GUS staining in the callus cells harboring the reporter gene sp38(∆266–289)–GUS and sp38+31nt to assure the staining quality.

–––Thank you for your comments. Lower and upper panels of Fig. 1B show the qualitative and quantitative data of GUS activities, respectively. Both panels similarly show the activities of the reporter GUS protein. The data in the upper panel is more accurate than those in the lower panel. Because a result of staining has been reported in the previous paper, the lower panel of Fig. 1B was deleted and corresponding figure legend are revised (lines 481–483).

3) P6 L5: “UTRc(mut252–265)” It is not clear what number corresponds to the sp38. The number should be consistent within the manuscript. For example, numbers should be added in the bar illustrations of Fig. 2 and Fig, 4. Supplementary Fig. S2 shows sp38 to start from #253 while mutations for constructs shown in Fig. 2 start with #252 with no explanation. Furthermore the confusion is deepened by the fact that Teramura et al. 2012 defines the sp38 sequence to start from #254 in their Figure　1. This contradiction should be solved or explained.

–––'sp38' is CUACAAAAAA AUACUCAGGU UUCAGAUCAU UUUUCGAG. In the Supplementary Fig. S2, the region showing sp38 was incorrect (this also included the additional region in UTRb). We amended this to the correct one. This sequence starts at nt 253 and ends at nt 289. To determine the effect of sp38 for translation efficiency, we modified the sequence of sp38 (nt 253–289) and the region around sp38 by replacing discrete 10–14 nt regions with their complementary nucleotides. In the revised manuscript, 'nt 253–289' was added, and this was changed as follows: To determine whether the sp38 nucleotide sequence determines UTRc translation efficiency, we modified this sequence inside sp38 (nt 253–289) and the region around sp38 by replacing discrete 10–14 nt regions with their complementary nucleotides [UTRc(mut252–265)–GUS, UTRc(mut266–275)–GUS, UTRc(mut276–285)–GUS, UTRc(mut286–295)–GUS, and UTRc(mut296–305)–GUS] (Fig. 2) (lines 123–127). Figs 2 and 4 were revised to clearly indicate sp38 in UTRc.

4) P7 L8: “starting from the nucleotide 330” Does this indicate that the first ORF starts from #330, whereas the first AUG is at #331? Alternatively does this indicate that the second half of UTRc starts from #330? The expression is misleading and must be revised.

–––According to the reviewer’s suggestion, we revised the text. This sentence was divided into two parts and 330 was amended to 331. Revised manuscript was as following: UTRc has three uORFs (uORF1, uORF2, and uORF3) located within the 5' UTRc second half. uORF1 and uORF2 start from the nucleotide 331 and 474 (Fig. 4; Supplementary Fig. S2), with the AUGs of uORF1 and uORF2 residing in a moderate initiation context with R (G and A, respectively) at position –3 with respect to the first nucleotide of the start codon (Kozak, 1986) that is recognized efficiently in plants (von Arnim et al. 2014) (lines 154–158). 

5) P9 L10: “deletion of the corresponding 18 nt” This should be written out in numbers.

–––According to the reviewer’s suggestion, we added the number of the nucleotides as "nt 511–529" (lines 209–210).

6) Figures: The resolution of the figures should be improved (ex, Fig. 1, Fig 5, Fig. 6).

–––According to the reviewer’s suggestion, we revised and improved the figures.

7) Typos: Supplementary Fig. S2. has “Fig4” written on the bottom.

–––According to the reviewer’s suggestion, we amended it.

8) Supplementary Fig. S3.: The right gel seems to be a image of two separately run gels. The image can be used to show single bands but cannot be used to describe the length difference noted in the text due to the truncation. The data should be revised for this purpose.

–––According to the reviewer’s suggestion, we added a sentence "A Gap is introduced between the lanes of the same gel." In addition, according to the comment of the editorial office, we uploaded the row data of the electrophoresis that indicates the precise sizes of the bands (lines 562–563).

---

## [Decision Letter · Decision Letter 1]

7 Jun 2021

Dissection of a rice OsMac1 mRNA 5' UTR to uncover regulatory elements that are responsible for its efficient translation

PONE-D-21-09082R1

Dear Dr. Shimada,

We’re pleased to inform you that your manuscript has been judged scientifically suitable for publication and will be formally accepted for publication once it meets all outstanding technical requirements.

Kind regards,

Minami Matsui

Academic Editor

PLOS ONE

Additional Editor Comments (optional):

Reviewers' comments:

Reviewer's Responses to Questions

**Comments to the Author**

1. If the authors have adequately addressed your comments raised in a previous round of review and you feel that this manuscript is now acceptable for publication, you may indicate that here to bypass the “Comments to the Author” section, enter your conflict of interest statement in the “Confidential to Editor” section, and submit your "Accept" recommendation.

Reviewer #1: All comments have been addressed

Reviewer #2: All comments have been addressed

2. Is the manuscript technically sound, and do the data support the conclusions?

Reviewer #1: Yes

Reviewer #2: Yes

3. Has the statistical analysis been performed appropriately and rigorously? 

Reviewer #1: Yes

Reviewer #2: Yes

4. Have the authors made all data underlying the findings in their manuscript fully available?

Reviewer #1: (No Response)

Reviewer #2: Yes

5. Is the manuscript presented in an intelligible fashion and written in standard English?

Reviewer #1: Yes

Reviewer #2: Yes

6. Review Comments to the Author

Reviewer #1: (No Response)

Reviewer #2: The manuscript titled “Dissection of a rice OsMac1 mRNA 5' UTR to uncover regulatory elements that are responsible for its efficient translation" has been revised to meet the standards of the Journal.

7. PLOS authors have the option to publish the peer review history of their article (what does this mean?). If published, this will include your full peer review and any attached files.

Reviewer #1: No

Reviewer #2: No

---

## [Editor Report · Acceptance letter]

1 Jul 2021

PONE-D-21-09082R1 

Dissection of a rice *OsMac1* mRNA 5' UTR to uncover regulatory elements that are responsible for its efficient translation 

Dear Dr. Shimada:

I'm pleased to inform you that your manuscript has been deemed suitable for publication in PLOS ONE. Congratulations! Your manuscript is now with our production department. 

Kind regards, 

on behalf of

Dr. Minami Matsui 

Academic Editor

PLOS ONE